# Phytochemical Screening, and In Vitro Evaluation of the Antioxidant and Dermocosmetic Activities of Four Moroccan Plants: *Halimium antiatlanticum*, *Adenocarpus artemisiifolius*, *Pistacia lentiscus* and *Leonotis nepetifolia*

Hicham Mechqoq [1], Sohaib Hourfane [2], Mohamed El Yaagoubi [1], Abdallah El Hamdaoui [1], Fouad Msanda [1], Jackson Roberto Guedes da Silva Almeida [3], Joao Miguel Rocha [4,5,*] and Noureddine El Aouad [2,*]

1 Laboratory of Biotechnologies and Valorization of Natural Resources, Faculty of Sciences, University Ibn Zohr, Agadir 80000, Morocco
2 Research Team on Natural Products Chemistry and Smart Technologies (NPC-ST), Polydisciplinary Faculty of Larache, University Abdelmalek Essaadi, Tetouan 93000, Morocco
3 Center for Studies and Research of Medicinal Plants (NEPLAME), Federal University of Vale do São Francisco, Petrolina 56304-205, Brazil
4 LEPABE—Laboratory for Process Engineering, Environment, Biotechnology and Energy, Faculty of Engineering, University of Porto, Rua Dr. Roberto Frias, 4200-465 Porto, Portugal
5 ALiCE—Associate Laboratory in Chemical Engineering, Faculty of Engineering, University of Porto, Rua Dr. Roberto Frias, 4200-465 Porto, Portugal
* Correspondence: jmfrocha@fe.up.pt (J.M.R.); n.elaouad@uae.ac.ma (N.E.A.)

**Abstract:** In this study, four Moroccan plants, *Halimium antiatlanticum*, *Adenocarpus artemisiifolius*, *Pistacia lentiscus* and *Leonotis nepetifolia*, were evaluated for their phytoconstituents and biological activities. Methanolic extracts of these plants were obtained by Soxhlet apparatus, phytochemical screening was performed, and the total phenolic and flavonoid contents were determined. Then, the antioxidant and dermocosmetic activities of the methanolic extracts were evaluated. The obtained results revealed that the leaves and/or aerial parts contained tannins, polyphenols, flavonoids, coumarins, carotenoids, terpenoids and saponins. The higher total phenolic content values were recorded on *Pistacia lentiscus* and *Halimium antiatlanticum* with 396.64 ± 30.79 and 304.96 ± 55.61 mgGAE/gDW, respectively. The antioxidant activity was measured by DPPH, ABTS and FRAP assays, and showed that *Pistacia lentiscus* and *Halimium antiatlanticum* were the most active extracts, with, respectively, IC50 values of 3.705 ± 0.445 and 5.037 ± 0.122 µg/mL for DPPH. The same results were observed for the FRAP and ABTS assays. Those extracts also showed a strong collagenase inhibitory activity at 200 µg/mL, with 78.51 ± 2.27% for *Pistacia lentiscus* and 73.10 ± 8.52% for *Halimium antiatlanticum*. *Adenocarpus artemisiifolius* showed the highest elastase inhibition rate, with 76.30 ± 5.29%. This study disclosed the dermocosmetic potential of *Halimium antiatlanticum* and *Adenocarpus artemisiifolius*, two Moroccan endemic plants that can be traditionally used by local populations or exploited by the cosmetic industry.

**Keywords:** phytochemical screening; polyphenol; flavonoid; antioxidant activity; dermocosmetic proprieties; collagenase; elastase

## 1. Introduction

Aromatic and medicinal plants (PAMs) are considered an inexhaustible source of bioactive natural substances. These substances have different chemical structures and plenty of biological properties, which makes them very coveted by the food, nutraceutical, pharmaceutical and cosmetic industries [1,2]. The diversity of natural substances in these plants is the result of different processes of evolution and adaptation to specific biological targets. Recent statistics estimate that active natural substances represent 55% of the cosmetic products on the market [3]. The global natural-based cosmetic products market was valued at approximately 22 billion USD in 2021 [4].

Nowadays, increasing awareness concerning natural products' low side effects has increased consumer demand for plant-based cosmetics, which has helped the rapid rising of this sector [5]. Those conditions have led to the emergence of many phytochemical and pharmaceutical investigations that have supported the use of natural products as active agents or ingredients for the elaboration of new products commonly known as cosmeceuticals [6,7]. Such products are hybrids of cosmetic and pharmaceutical compounds and possesses both therapeutic and beautification potentials.

The Moroccan kingdom, located in the northwest of Africa, has on its territory some of the most original flora, with more than 5200 species and subspecies spread over 155 families, and an endemic rate of 20% [8]. This plant biodiversity is mainly due to its particular geographical location and different climates. The Moroccan climate is generally Mediterranean characterized by a dry summer and rainy winter. The presence of the Mediterranean Sea in the North and the Atlantic Ocean in the west induces a climate variation, thus dividing the country into several bioclimatic floors.

This paper investigated the chemical composition and potential antioxidant and dermocosmetic proprieties of four plant species present in Morocco (Figure 1). *Halimium antiatlanticum* and *Adenocarpus artemisiifolius* are two Moroccan endemic plant species that belong to the Cistaceae and Fabaceae families, respectively. Those two plants grow spontaneously in the semi-arid central areas of Morocco, and to date, no ethnobotanical or phytochemical studies have been carried out on those two species. *Pistacia lentiscus* and *Leonotis nepetifolia* have not been studied either. These two ubiquitous plants species belong to the Anacardiaceae and Lamiaceae families, respectively. In addition to their spontaneous growing in the Central and North areas of Morocco, they are also found in other geographical areas. Indeed, *Pistacia lentiscus* is largely distributed on the Mediterranean area, whereas *Leonotis nepetifolia* is present on both the African and American continents, where it is considered an invasive species.

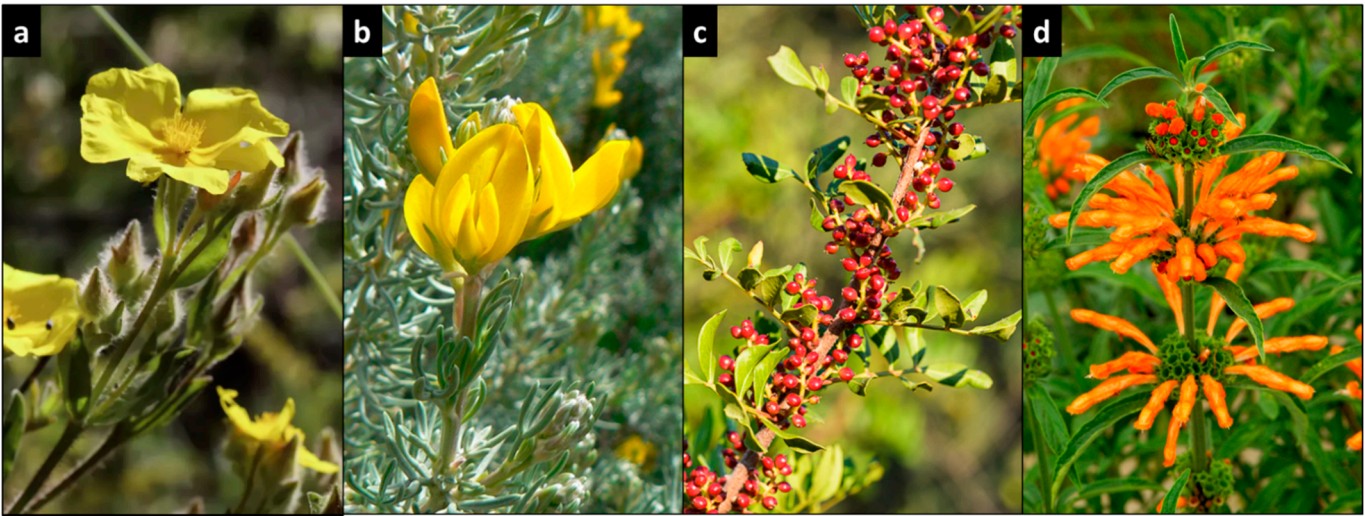

**Figure 1.** General aspect of Halimium antiatlanticum (**a**), Adenocarpus artemisiifolius (**b**), pistacia lentiscus (**c**), and Leonotis nepetifolia (**d**) aerial parts.

## 2. Material and Methods

### 2.1. Chemicals and Reagents

Petroleum ether, methanol and dichloromethane were acquired from Sigma Aldrich (Paris, France). Hydrochloric, glacial acetic and sulfuric acids were purchased from Panreac Quimica SA (Barcelona, Spain). Dragendorf, mayer, Folin–Ciocalteu reagents, phosphate-buffered saline (PBS) and trichloroacetic acid (TCA) were obtained from Alfa Aesar GmbH & Co. (Germany). Ferric chloride, potassium hydroxide, sodium hydroxide, sodium carbonate, sodium nitrite, aluminum chloride, potassium ferricyanide, potassium pertsulfate, 2,2-diphenyl-1-picrylhydrazyl (DPPH) and 2,2'-azino-bis(3-ethylbenzothiazoline-6-sulfonic

acid (ABTS) were acquired from Merck (Darmstadt, Germany). Iron III chloride, gallic acid, quercetin and ascorbic acid were purchased from Sigma Aldrich (Paris, France). For collagenase inhibition assay, collagenase from *Clostridium histolyticum* (≥1 FALGPA units/mg; EC number: 232-582-9), fluorogenic substrate peptide MMP-2 (MCA-Pro-Leu-Ala-Nva-DNP-Dap-Ala-Arg-NH2) and chlorhexidine were obtained from Sigma-Aldrich. For elastase inhibition assay, porcine pancreatic elastase type IV (≥4 units/mg protein; EC Number 254-453-6), N-succinyl-Ala-Ala-Ala-p-nitroanilide (EC Number 257-823-5), tris(hydroxymethyl)aminomethane hydrochloride (Trizma-HCl) and elastatinal were purchased from Sigma-Aldrich.

## 2.2. Plant Material

Plants were collected between April–May 2018 in areas between Ida-ou-Tnane mountain and Agadir (Table 1). These plants were identified on the field by Prof. Fouad Msanda, Faculty of Sciences, University Ibn Zohr-Agadir. The aerial parts and/or leaves were washed thoroughly and dried in shade. Then, they were grinded to a fine powder using a laboratory grinding mill (Polymix PX-MFC 90D, Switzerland) and stored in glass bottles in the dark at 4 °C.

**Table 1.** Botanical name, family, site of collection and parts used of plant species tested in this study.

| Botanical Name | Family | Site | Used Parts |
|---|---|---|---|
| *Halimium antiatlanticum* Maire and Wilczek | Cistaceae | Souss-Massa Draa valley | Leaves + stems |
| *Adenocarpus artemisiifolius* Jahandiez, Maire and Weiller | Fabaceae | Ida-ou-Tanane mountain | Leaves + stems |
| *Pistacia lentiscus* (L.) | Anacardiaceae | Agadir | Leaves |
| *Leonotis nepetifolia* (L.) R.Br. | Lamiaceae | Agadir | Leaves |

## 2.3. Extraction and Sample Preparation

The different plants materials were extracted by Soxhlet (Labbox, Chartres, France). For this, 50 g of each material was pre-extracted with petroleum ether in order to eliminate non-polar compounds. Then, the plant materials were extracted with methanol. The obtained extracts were filtered through whatman No. 1 (11 μm-φ, cellulose filter) using vacuum filtration in an Erlenmeyer; the obtained extracts were concentrated under vacuum at 40 °C, dried under nitrogen, and stored in sealed dark glass vials at 4 °C. Each extraction was performed in triplicate.

## 2.4. Determination of the Extraction Yields

The extraction yields of the different plant extracts were calculated using Equation (1) below:

$$\text{Yield of Extraction (\%)} = \frac{\text{Extract weight (g)}}{\text{Plant material weight (g)}} \times 100 \qquad (1)$$

## 2.5. Phytochemical Screening

2.5.1. Qualitative Analysis on Phytochemical Constituents

Phytochemical screening of the extracts was performed in order to detect the eventual presence of different secondary metabolites, such as alkaloids, tannins, polyphenols, flavonoids, anthocyanidins, terpenoids, anthraquinons, carotenoids and saponins, following the protocols described below.

For alkaloids, two tests were performed. The first was the Mayer test, in which 0.2 mL of extract were added to 5 mL of aqueous hydrochloric acid (1%) on a steam bath. Then, 1 mL of the filtrate (Whatman paper No 1, 11 μm-φ, cellulose filter) was taken, and 3 drops of Mayer's reagent were added. The formation of creamy yellow/white precipitate indicates the presence of alkaloids [9]. Second, the Dragendorff test, in which 0.2 mL of extract was added to 5 mL of aqueous hydrochloric acid (1%, *v/v*) in a steam bath. Then,

3 drops of Dragendorff's reagent were added. The formation of reddish brown precipitate indicates the presence of alkaloids [9].

For tannins, 0.2 mL of the extract was boiled in 10 mL of water in a test tube and then filtered (Whatman paper No 1, 11 μm-$\phi$, cellulose filter). Afterwards, a few drops of 0.1% ($w/v$) ferric chloride were added. The appearance of blackish blue color indicates the presence of tannins [10]. For polyphenols, 0.5 mL of the extract was added to 5 mL of distilled water in a test tube. A few drops of 5% ferric chloride were added. A bluish black shows the presence of polyphenols [11]. Flavonoids have been detected using the alkaline reagent test. This test consists of 0.5 mL of extract with the addition of a few drops of sodium hydroxide. An intense yellow color that disappears after adding dilute HCl indicates the presence of flavonoids [11]. For anthocyanidins, an extract volume of 0.5 mL was added to aqueous KOH 10% ($v/v$). A red color suggests the presence of anthocyanidins [12]. Moreover, the Salkowski test was performed for terpenoids. To that purpose, 0.2 mL of each extract was diluted on 2 mL of chloroform. Then, 3 mL of concentrated $H_2SO_4$ (95–97%, $v/v$) was carefully added to form 2 layers. A reddish-brown color on the interphase shows the presence of terpenoids [10,11]. Anthraquinons were assayed with the potassium hydroxide test. For this test, 0.5 mL of extract was added to 0.1 mL of aqueous KOH solution (10%, $w/v$). A red color indicates the presence of anthraquinons [13]. Furthermore, carotenoids were assayed by preparing 0.5 mL of extract with 1 mL of commercial HCl (37%, $v/v$). A blue-green color suggests the presence of carotenoids [12]. Finally, saponins were detected using the Froth test: 0.1 mL of aqueous extract was mixed with 1.5 mL of distilled water in a test tube. The solution was shaken vigorously and left for 20 min. The persistence of a 1 cm foam for 10 min indicates the presence of saponins [14].

2.5.2. Quantitative Analysis on Phytochemical Constituents

The total content of polyphenols was determined using the Folin–Ciocalteu method. The reagent consists of a mixture of phosphotungstic acid ($H_3PW_{12}O_{40}$) and phosphomolybdic acid ($H_3PMo_{12}O_{40}$). The oxidation of the phenols reduces this reagent to a mixture of blue oxides of tungsten and molybdenum. The intensity of the color is proportional to the level of oxidized phenolic compounds, whose maximum absorbance is 760 nm [15]. The extracts were diluted to prepare 1 mg/mL solutions. Then, 100 μL of this dilution was mixed with 2 mL of $Na_2CO_3$ solution (2%, $w/v$). After 5 min of incubation, 100 μL of Folin–Ciocalteu reagent was added. The mixture was incubated in darkness for 60 min, and afterwards, the absorbance was measured at 700 nm using a spectrophotometer (Zuzi, model 4201/50, Navarra, Spain). Gallic acid was used as a standard for the calibration curve, and the results were expressed as mg of gallic acid equivalent per gram of dry weight (mg GAE/g DW) [16].

The total content of flavonoids was determined using the colorimetric aluminum trichloride method. In this method, aluminum trichloride ($AlCl_3$) forms a yellow complex with flavonoids, and the addition of sodium hydroxide (NaOH) gives a pink-colored complex that absorbs at 510 nm [12,16]. For that, 500 μL of the extracts were mixed with 2 mL of distillated water, 0.15 mL of $NaNO_2$ solution (5%, $w/v$) and 0.15 mL of $AlCl_3$ solution (10%, $w/v$). After 5 min, 2 mL of NaOH solution (4%, $w/v$) and 0.2 mL of distilled water were added to the mixture. The absorbance was measured at a wavelength of 510 nm. Quercetin was used as a standard for the calibration curve, and the results were expressed as milligrams of quercetin equivalent per gram of dry weight (mg QE/g DW) [17].

*2.6. Antioxidant Activity*

2.6.1. DPPH Radical Scavenging Activity

The free radical scavenging activity was measured by DPPH radical. The DPPH (2,2-diphenyl-1-picrylhydrazyl) uses a relatively stable radical that has a violet color. The antioxidants present reduce this radical to a yellow compound (diphenyl picryl hydrazine). The percentage of inhibition of DPPH is calculated by measuring the absorbance at 517 nm [18].

The extracts were diluted at various concentrations (50, 250, 350, 500, 750 and 1000 µg/mL). Fifty microliters of each concentration was added to 1950 µL of DPPH solution in methanol (25 mg/L). After incubation in the dark for 30 min, the maximum absorbance was measured at 515 nm using a spectrophotometer (Zuzi, model 4201/50, Navarra, Spain). The radical-scavenging activities of the tested extracts, expressed as percentage inhibition of DPPH, were calculated according to Equation (2):

$$\text{DPPH inhibition (\%)} = \frac{\left(A_{blank} - A_{sample}\right)}{\left(A_{blank}\right)} \times 100 \qquad (2)$$

where $A_{blank}$ is the absorbance of the control reaction (containing all reagents except the tested compound), and $A_{sample}$ is the absorbance of the test compound. The percentage of inhibition was plotted against concentration, and the linear regression equation was used to calculate the IC50 value. Ascorbic acid was used as a positive control, and all tests were carried out in triplicate. Low IC50 values indicate the greatest antioxidant activity.

### 2.6.2. Ferric Reducing Antioxidant Power (FRAP) Assay

The FRAP method is based on the reduction of ferric ion ($Fe^{3+}$) present in the potassium ferricyanide (III) [$K_3Fe(CN)_6$] complex to ferrous ion ($Fe^{2+}$). The reaction is monitored by measuring the absorbance at 700 nm [19]. The total antioxidant activity by ferric thiocyanate (FTC) was performed using the method described by Oyaizu [20], with some modifications. Briefly, dilutions of extracts were prepared at various concentrations (50, 250, 350, 500, 750 and 1000 µg/mL). Then, 0.1 mL of plant extract was mixed with 2.5 mL of PBS (0.04 M; pH 7.4), and 2.5 mL of 1% (*w/v*) solution of potassium ferricyanide. The mixture was incubated at 50 °C for 20 min. After incubation, 2.5 mL of a 10% (*w/v*) trichloroacetic acid (TCA) solution was added. Then, 2.5 mL of the supernatant of each sample and concentration was mixed with 2.5 mL of distilled water and 0.5 mL of 0.1% iron chloride (III) ($FeCl_3$). For this assay, the absorbance was read at 700 nm. Ascorbic acid was used as a positive control. Total antioxidant activity was calculated as percentage of inhibition, using the same equation previously used for DPPH scavenging activity (Equation (2)). An efficient concentration of 50% (EC50) was determined by plotting the inhibition values against concentration. All tests were carried out in triplicate. Low EC50 values indicate the highest antioxidant activity.

### 2.6.3. ABTS Radical Scavenging Activity

The ABTS radical cation scavenging activity was evaluated by a slight modified experimental protocol of Re, et al. [21]. For this, an ABTS solution (7 mM) with potassium persulfate (2.45 mM) solution was prepared and kept overnight in the dark to obtain a dark solution containing ABTS radical cations. A volume of the ABTS radical cation solution was diluted with 50% (*v/v*) methanol, with an initial absorbance value of 0.700 ± 0.02 at 745 nm. For the free radical scavenging activity of the samples, 300 µL of each sample was mixed with 3.0 mL of ABTS solution. The absorbance value was measured one minute after mixing the solution, then rechecked up to 5 min. The percentage of inhibition was calculated according to Equation (3):

$$\text{Scavenging effect (\%)} = \frac{\left(A_{blank} - A_{sample}\right)}{\left(A_{blank}\right)} \times 100 \qquad (3)$$

where $A_{blank}$ is the absorbance of the control reaction (containing all reagents except the tested compound), and $A_{sample}$ is the absorbance of the test compound. The antioxidant capacity of test samples was expressed as EC50, i.e., the concentration necessary for 50% reduction of ABTS. Ascorbic acid was used as a positive control, and all tests were carried out in triplicate.

*2.7. Dermocosmetic Activities*

2.7.1. Collagenase Inhibition Activity

The collagenase inhibitory potential was determined using the spectrofluorimetric method previously described by Mechqoq, Hourfane, El Yaagoubi, El Hamdaoui, da Silva Almeida, Rocha and El Aouad [7]. In this experimental assay, a fluorogenic substrate (metalloproteinase-2) is degraded by collagenase producing fluorescent signal. For this assay, collagenase from *Clostridium histolyticum* (EC number: 232–582–9; Sigma-Aldrich) was prepared in Trizma-base buffer (10 mM, pH = 7.3). In a 96-well microplate, 120 µL of buffer, 40 µL of sample and 40 µL of collagenase enzyme solution (60 µg/mL) were incubated at 37 °C for 10 min. Then, 40 µL of the fluorogenic substrate was added, and the mixture was incubated at 37 °C for 30 min. The fluorescent intensity was measured at excitation and at emission wavelengths of 320 and 405 nm, respectively, using an Infinite 200 PRO series (Tecan, Männedorf, Switzerland) plate reader. The samples were evaluated at 250 µg/mL, and the extracts that exhibited an inhibition percentage above 70% were then evaluated for lower concentrations in order to calculate the IC50 values. Chlorhexidine was used as a positive control of the assay (IC50 = 50 µM) [16]. Experiments were performed in three analytical replicates. The inhibition percentage of collagenase was calculated as described in Equation (4):

$$\text{Collagenase inhibition } (\%) = \frac{(A - B) - (C - D)}{(A - B)} \times 100 \tag{4}$$

where A is the control (without (*w/o*) sample), B the blank (*w/o* sample, *w/o* collagenase), C the sample and D the blank sample (*w/o* collagenase).

2.7.2. Elastase Inhibition Activity

Elastase inhibition activity was evaluated according to a method previously described by Angelis, et al. [22], using N-succinyl-Ala-Ala-Ala-p-nitroanilide (Sigma-Aldrich, EC Number 257–823–5) as a substrate. Furthermore, the substrate degradation causes the release of p-nitroaniline, which can be monitored spectrophotometrically at 405 nm. Porcine pancreatic elastase type IV ($\geq$4 units/mg protein; EC Number 254–453–6, Sigma-Aldrich) and *N-succinyl-Ala-Ala-Ala-p-nitroanilide* were dissolved in Trizma-base buffer (50 mM, pH = 7.5). Thus, 70 µL of Trizma-base buffer, 10 µL of the samples and 5 µL of elastase (0.45 U/mL) were mixed and incubated in a 96-well microplate for 10 min in the dark. Then, 20 µL of substrate (2 mM) was added in each well, and the plate was incubated for 30 min. The absorbance was measured at 405 nm with an Infinite 200 PRO (Tecan) plate reader. The samples were evaluated at 200 µg/mL. Then, the most promising ones were evaluated at lower concentrations to determinate the IC50 values. As a strong competitive inhibitor of elastase (IC50 = 0.5 µg/mL), elastatinal was used as a positive control. The experiments were performed in triplicate (3 analytical replicates). The inhibition percentage of elastase was calculated by the following expression of Equation (5):

$$\text{Elastase inhibition } (\%) = \frac{(A - B) - (C - D)}{(A - B)} \times 100 \tag{5}$$

where A is the control (without (*w/o*) sample), B the blank (*w/o* sample, *w/o* elastase), C the sample and D the blank sample (*w/o* elastase).

*2.8. Statistical Analysis*

All experimental data were expressed as mean $\pm$ standard deviation by measuring three independent replicates. Means were compared statistically using the Statistica® software v6.1, (Statsoft, Inc., Palo Alto, CA, USA), with Student's *t* test (significance level $p < 0.05$).

## 3. Results and Discussion

### 3.1. Extraction Yields

The extraction of the powders of aerial parts and leaves was performed using two different solvents, petroleum ether and methanol. The plant material was defatted with petroleum ether and then extracted with methanol. Table 2 presents the different extracts aspects and the extraction yields.

**Table 2.** Aspect and extraction yield of the studied plant extracts.

| Botanical Name | Extract Color | Extraction Yield (%) |
|---|:---:|:---:|
| *Halimium antiatlanticum*, Maire and Wilczek | Green | 34.21 |
| *Adenocarpus artemisiifolius* Jahandiez, Maire and Weiller | Red-green | 27.54 |
| *Pistacia lentiscus* (L.) | Green | 46.35 |
| *Leonotis nepetifolia* (L.) R.Br. | Dark green | 13.13 |

The *Pistacia lentiscus* and *Halimium antiatlanticum* methanolic extracts were in the form of a green powder with a rough aspect, whereas *Adenocarpus artemisiifolius* and *Leonotis nepetifolia* led to red-green and dark-green powders with a shiny aspect. The highest yield value was obtained by *Pistacia lentiscus* methanolic extract with 46.35%, followed by *Halimium antiatlanticum* and *Adenocarpus artemisiifolius* methanolic extracts with 34.21 and 27.54%, respectively. The lowest yield value was obtained with *Leonotis nepetifolia* (13.13%).

In a study conducted by Gardeli, et al. [23], the methanolic maceration of *Pistacia lentiscus* leaves presented a yield value of 45.2%. In another paper, the authors reported the use of microwave-assisted extraction on *Pistacia lentiscus* leaves. This method increased the yield to 47.98% [24]. Concerning *Leonotis nepetifolia*, Oliveira, et al. [25] described the hydro-ethanolic maceration of the leaves, which yielded a value of 15.69%. Moreover, *Halimium antiatlanticum* methanolic extract gave a yield value of 32.6% according to Talibi, et al. [26]. The yield value of *Adenocarpus artemisiifolius* has not been reported in the literature.

A comparison of the bibliographic data shows that the reported yield values are relatively similar to our findings, which may suggest that the extraction procedure allowed an exhaustive extraction, with the recovery of a quantity of products similar to the ones cited in the literature.

### 3.2. Phytochemical Screening

3.2.1. Qualitative Analysis on Phytochemical Constituents

The methanolic extracts of the different plant species were diluted and prepared in test tubes and submitted to a qualitative phytochemical screening. Table 3 presents the results of the qualitative phytochemical screening.

**Table 3.** Results of the qualitative analysis on phytochemical constituents.

| Phytochemical Constituents | *Halimium antiatlanticum* Maire and Wilczek | *Adenocarpus artemisiifolius* Jahandiez, Maire and Weiller | *Pistacia lentiscus* (L.) | *Leonotis nepetifolia* (L.) R.Br. |
|---|:---:|:---:|:---:|:---:|
| Alkaloids | − | +++ | − | − |
| Tannins | +++ | ++ | +++ | ++ |
| Polyphenols | +++ | ++ | +++ | + |
| Flavonoids | ++ | + | ++ | + |
| Anthocyanidins | ++ | + | + | − |
| Terpenoids | +++ | +++ | +++ | − |
| Anthraquinons | − | ++ | − | − |
| Carotenoids | − | ++ | − | + |
| Saponins | ++ | ++ | ++ | + |

+++ Strong positive test ++ Low positive test + weak positive test − Negative test.

The preliminary phytochemical screening of the *Halimium antiatlanticum* methanolic extract revealed the presence of different phytoconstituents, with an abundance of tannins, polyphenols and terpenoids. This extract also contained smaller amounts of flavonoids, anthocyanidins and saponins. Regarding the *Pistacia lentiscus* methanolic extracts, the results were similar to those of *Halimium antiatlanticum*, with slight variation. Finally, the *Pistacia lentiscus* extract displayed a lower concentration of anthocyanidins.

*Adenocarpus artemisiifolius* methanolic extract disclosed different results, with high levels of alkaloids and terpenoids, and smaller amounts of tannins, polyphenols, anthraquinons, carotenoids and saponins. The lowest phytochemical screening results were observed with *Leonotis nepetifolia* methanolic extract. Indeed, the qualitative screening also showed the presence of tannins and scarcer amounts of polyphenols, flavonoids, carotenoids and saponins.

These results are in line with what has been demonstrated in previous phytochemical investigations. According to those studies, the screening of methanolic extracts of *Pistacia lentiscus* aerial parts proved the presence of tannins, flavonoids and saponins [27,28]. The phytochemical screening of a methanolic extract of *Leonotis nepetifolia* leaves reported by Trivedi, et al. [29] cited the presence of small amounts of polyphenols, tannins, flavonoids and saponins.

These findings support the postulate that these plants possess numerous interesting chemical constituents in different amounts (mainly polyphenols). These compounds may have pharmacological activities. In the current study, the authors were particularly interested in the polyphenols and flavonoids due to their large biological spectra.

### 3.2.2. Quantitative Analysis on Phytochemical Constituents

Due to their several dermocosmetic proprieties, we have chosen to focus our investigation on the polyphenols and flavonoids of the studied plant species. The results of the total phenolic and flavonoid content evaluations are summarized in Figure 2.

All the evaluated extracts contained a considerable quantity of phenolic compounds. The highest content of polyphenols was found in the methanolic extract of *Pistacia lentiscus* with $396.638 \pm 30.79$ mg GAE/g DW, followed by *Halimium antiatlanticum* and *Adenocarpus artemisiifolius* with, respectively, $304.96 \pm 82.81$ mg GAE/g DW and $201.687 \pm 13.533$ mg GAE/g DW. *Leonotis nepetifolia* showed the lowest phenolic content of $53.883 \pm 6.828$ mg GAE/g DW.

According to Sobolewska, et al. [30], *Leonotis nepetifolia* methanolic extract contained 6.4 mg GAE/g DW of polyphenols. More recent studies suggests that *Leonotis nepetifolia* have low levels of polyphenols [25,31]. Regarding *Pistacia lentiscus*, published works suggest that the methanolic extracts of this plant contain high levels of phenolic contents, with values ranging between 480 and 630 mg GAE/g DW [23,32].

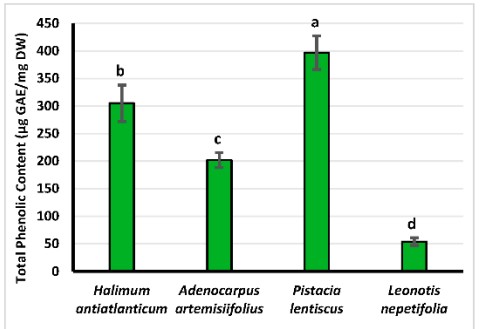 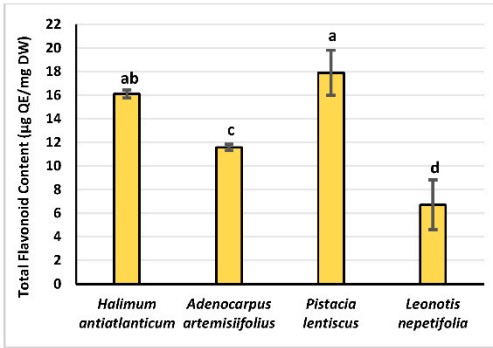

**Figure 2.** Total phenolic and flavonoid contents of the studied plants species (a–d represent significant differences at $p < 0.05$ by Newman–Keuls test).

Concerning the total flavonoid contents, all the evaluated extracts contained smaller amounts of flavonoids in contrast to the phenolic compounds. Those results revealed that

the methanolic extracts of *Pistacia lentiscus* and *Halimium antiatlanticum* had the highest flavonoid contents of 17.894 ± 1.911 and 16.105 ± 0.33 mg QE/g DW, respectively, followed by *Adenocarpus artemisiifolius*, with 11.569 ± 0.26 mg QE/g DW, and, finally, *Leonotis nepetifolia*, with 6.7 ± 2.105 mg QE/g DW.

In a recent investigation on the phytochemistry of *Leonotis nepetifolia* conducted by Imran, Suradkar and Koche [31], its flavonoid content was evaluated. According to these authors, the hydro-methanolic extract contained 1.47 ± 0.11 mg QE/g DW of flavonoids. Despite being very low, this result is similar to the one obtained in our current study. Many publications reported flavonoid contents of different *Pistacia lentiscus* parts, describing high amounts of flavonoids of 67.48 mg QE/g DW for the Soxhlet aqueous extract, and 38.7 mg QE/g DW for the hydro-methanolic extract [24,32].

The originality of this study comes from the fact that the chemical evaluation of *Halimium antiatlanticum* has not been studied so far. The only papers featuring *Halimium antiatlanticum* were a group of studies published between 2012 and 2013 [26,33,34]. In these manuscripts, organic extracts of this plant were evaluated for their antifungal activity and showed very interesting results. For *Adenocarpus artemisiifolius*, the only published paper to date is a chemical comparative study conducted by Essokne, et al. [35], where the flavonoids of 10 plants belonging to the *Adenocarpus* genus were analyzed using high-performance liquid chromatography with diode array detection and mass spectrometry (HPLC-DAD-MS). The results of this study revealed the presence of flavones and isoflavones, represented by the flavone di-C-glycosides, flavone mono-C-glycosides, flavone 4′-O-glycosides, 5-OH-isoflavone O-glycosides and 5-Deoxy- and 5-OMe-isoflavone O-glycosides.

*3.3. Antioxidant Activity*

To compare the antioxidant capacity of each methanolic extract, a single test seems not to be enough, since the antioxidant activity involves various mechanisms of action. This suggests that the evaluation of the antioxidant activity of the extracts should be undertaken by different techniques, representing two different antioxidant mechanisms [36]. In agreement, the antioxidant activity was investigated using free radical scavenging (DPPH and ABTS), in addition to reducing antioxidant power (FRAP) methods. Table 4 summarizes the results of the different antioxidant assays.

The different extracts displayed variable DPPH inhibition intensities. The extract efficiencies are determined by calculating its IC50 (µg/mL), keeping in mind that the smaller the values of IC50, the better. The ascorbic acid used as a standard for comparison demonstrated the lowest IC50 value, with 0.836 ± 0.017 µg/mL. The nearest IC50 value was observed with *Pistacia lentiscus* (3.705 ± 0.445 µg/mL), followed by *Halimium antiatlanticum*, with 5.037 ± 0.122 µg/mL, then *Adenocarpus artemisiifolius* and, finally, *Leonotis nepetifolia*, with 58.813 ± 8.4 µg/mL and 779.407 ± 101.57 µg/mL, respectively.

**Table 4.** Antioxidant assays results in IC50 and EC50 (mean values ± standard deviation) (a–e represent significant differences at $p < 0.05$ by Newman–Keuls test).

| Control/Species | DPPH $IC_{50}$ (µg/mL) | FRAP $EC_{50}$ (µg/mL) | ABTS $EC_{50}$ (µg/mL) |
|---|---|---|---|
| Ascorbic acid | 0.836 ± 0.017 [e] | 32.36 ± 1.306 [e] | 2.831 ± 0.52 [e] |
| *Halimium antiatlanticum* Maire and Wilczek | 5.037 ± 0.122 [c] | 71.613 ± 1.23 [c] | 10.98 ± 0.122 [c] |
| *Adenocarpus artemisiifolius* Jahandiez, Maire and Weiller | 58.813 ± 4.4 [b] | 420.88 ± 5.39 [b] | 67.1 ± 10.75 [b] |
| *Pistacia lentiscus* (L.) | 3.705 ± 0.445 [d] | 65.63 ± 1.41 [d] | 3.285 ± 0.911 [d] |
| *Leonotis nepetifolia* (L.) R.Br. | 779.407 ± 10.57 [a] | 597.96 ± 20.06 [a] | 582.3 ± 37.7 [a] |

The ferric reducing antioxidant power is evaluated by calculating the EC50 values. Higher reducing power values are associated with lower EC50 values. Ascorbic acid

exhibited excellent activity, with EC50 values of 2.36 ± 1.306 μg/mL. *Pistacia lentiscus* extract had the stronger activity, with an EC50 value of 6.63 ± 1.41 μg/mL, followed by *Halimium antiatlanticum* and *Adenocarpus artemisiifolius*, with 7.613 ± 1.23 μg/mL and 42.88 ± 5.39 μg/mL, respectively. The lowest reducing power was detected for *Leonotis nepetifolia* extract, with 597.96 ± 20.06 μg/mL.

The ABTS scavenging activity of the evaluated extracts indicated variable intensities depending on the extracts. The lowest EC50 value was observed in *Pistacia lentiscus* and *Halimium antiatlanticum* with 3.285 ± 0.911 μg/mL and 10.98 ± 0.122 μg/mL, respectively. *Adenocarpus artemisiifolius* and *Leonotis nepetifolia* showed 67.1 ± 10.75 μg/mL and 582.3 ± 37.7 μg/mL, respectively. These results were higher than the standard, *viz. ascorbic acid*, with an EC50 value of 1.831 ± 0.52 μg/mL.

In a study published by Rebaya, et al. [37], organic and aqueous extracts of *Halimium halimifolium* were evaluated for their free radical scavenging against DPPH and ABTS. The ethanolic and aqueous extracts of this plant showed IC50 values of 1.19 μg/mL and 1.548 μg/mL, respectively, for DPPH trial, and 10.4 μg/mL and 16.7 μg/mL, respectively, for ABTS [37]. These values were lower than ours in the present study. According to Gardeli, Vassiliki, Athanasios, Kibouris and Komaitis [23], the methanolic extract of *Pistacia lentiscus* had an IC50 value of 5.09 ± 0.1 μg/mL. The antioxidant activity of *Pistacia lentiscus* with ABTS has not been recorded in the literature. However, other species of *Pistacia* have been evaluated for ABTS antioxidant test. According to Rigane, et al. [38], the ethanolic extract of *Pistacia atlantica* had an EC50 value of 42 ± 0.00 μg/mL. In another study published by Abu-Lafi, et al. [39], the hydro-alcoholic extract showed an EC50 value of 53.1 ± 6.6 μmol/g [39]. The antioxidant activity of *Leonotis nepetifolia* extracts has also been reported by Sobolewska, Paśko, Galanty, Makowska-Wąs, Padło and Wasilak [30]. In their study, the soxhlet methanolic extract showed very low activity (the IC50 was not mentioned). These previous investigations confirm the data obtained in our research. On the other hand, Takeda, et al. [40] investigated the antioxidant DPPH activity of *Leonotis nepetifolia*'s in purified fractions. These authors purified some compounds and found very low IC50 values with 0.008 μg/mL for martinoside, and 0.006 μg/mL for lavandulifolioside. These results are justified by the fact that the purified fractions are generally more active than the crude extracts.

Although the original FRAP assay uses tripyridyltriazine (TPTZ) as the iron-binding ligand, other alternative ligands have also been employed for ferric binding, such as potassium ferricyanide—which is the most popular ferric reagent used in FRAP assays [41]. The EC50 values of the TPTZ assay are expressed in mM $Fe^{2+}$/g, whereas the potassium ferricyanide method values are expressed in μg or mg per mL.

In the previously cited study conducted by Rebaya, Belghith, Baghdikian, Leddet, Mabrouki, Olivier, Cherif and Ayadi [37], the ethanolic and methanolic extracts of *Halimium halimifolium* showed values of 55.63 mM $Fe^{2+}$/g and 53.02 mM $Fe^{2+}$/g, respectively [37]. Another study realized by Sobolewska, Paśko, Galanty, Makowska-Wąs, Padło and Wasilak [30] on *Leonotis nepetifolia* reported that the methanolic extract obtained by hot extraction at solvent boiling temperature had an EC50 value of 43.25 ± 1.16 mM $Fe^{2+}$/kg. These two assays were carried out using the TPTZ method. Since our study involved the potassium ferricyanide method, a comparison of results with the previous cited bibliographic references is not possible [30].

The only reported work describing the Ferric reducing power of *Adenocarpus* species was published by Berber, et al. [42]. In this study, the methanolic extract of *Adenocarpus complicates* exhibited an EC50 value of 901.25 ± 13.97 μg/mL [42]. This value was higher than our results, which may be due to the variation of phenolic content between *Adenocarpus* species. For *Pistacia lentiscus*, two studies have reported the antioxidant activity with the FRAP potassium ferricyanide method. The first one was reported by Ghenima, et al. [43], in which the aqueous extract of *Pistacia lentiscus* leaves showed an EC50 value of 54.06 ± 12.66 μg/mL in comparison with the ascorbic acid 38.92 ± 3.16 μg/mL [43]. The second one was undergone by Hemma, et al. [44]. In this study, the methanolic extract of

leaves showed lower ferric reducing power with EC50 207 $\pm$ 0.2 µg/mL in contrast with the ascorbic acid 36 $\pm$ 1 µg/mL [44].

All the three analytical methods used in the current work for the evaluation of antioxidant activity DPPH, ABTS and FRAP are spectrophotometric.

DPPH was found to be the most preferred method for the determination of antioxidant activity. This method is often used and reported [45–50] because it is easy to set, rapidly performed and has high reproducibility [51]. In the present study, the antioxidant activity of each sample was measured three times to test the reproducibility of the assay. DPPH showed high reproducibility. The ABTS assay is widely reported for the evaluation of antioxidant capacity. Unlike the DPPH assay, ABTS is time-consuming because the ABTS radicals generation process takes around 12–16 h (reaction of ABTS with potassium persulphate). The major disadvantage of this this assay is that the ABTS radicals are not very stable, and the results have little reproducibility. However, this assay is very easy to set up, and ABTS is soluble in both water and organic solvents [41]. The FRAP assay is simple to set up and inexpensive. It is also less time-consuming in terms of preparing the chemicals of the working solution.

In our study the results obtained by FRAP are found to be reproducible for all concentrations. However, in terms of preparation, it is relatively hard to prepare in comparison with a DPPH assay. Indeed, the DPPH method may be considered the easiest and more accurate.

### 3.4. Dermocosmetic Activities

Collagen and elastin are two key elements of the skin extracellular matrix. They are the main responsible for the skin elasticity and firmness [52], and their enzymatic degradation by collagenase and elastase is one of the main causes of skin intrinsic aging [53]. Figure 3 regroups the results of the collagenase and elastase inhibition assays of the different plant extracts.

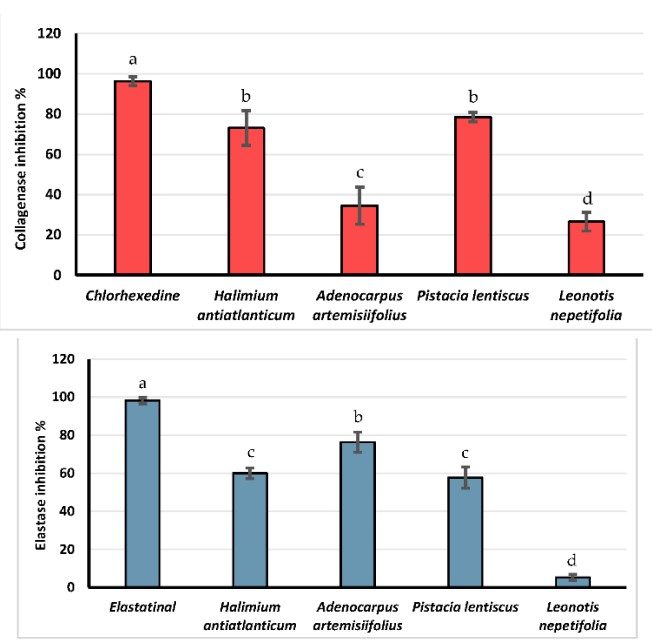

**Figure 3.** Collagenase and elastase inhibition capacities of the studied plants species (a–d represent significant differences at $p < 0.05$ by Newman–Keuls test).

The observation of the collagenase inhibition assay results (200 µg/mL) shows that the highest inhibition value was observed with the Chlorhexedine (positive control) with 96.33 $\pm$ 2.3%, followed by *Pistacia lentiscus* and *halimium antiatlanticum*, with 78.51 $\pm$ 2.27 and 73.10 $\pm$ 8.52%, respectively. The lowest inhibition values were attained with *Adenocarpus artemisiifolius* (34.53 $\pm$ 9.20%) and *Leonotis nepetifolia* (26.61 $\pm$ 4.69%). The evaluation of the elastase inhibition potential (300 µg/mL) of the different plant extracts showed very

interesting results. The highest inhibition activity was observed with *Adenocarpus artemisiifolius* extract, with an inhibition value 76.30 ± 5.29%, followed by *Halimium antiatlanticum* and *Pistacia lentiscus*, with 60.05 ± 2.76 and 57.70 ± 5.54%, respectively.

In a study published by Elloumi, et al. [54], the *Pistacia lentiscus* methanolic extract was evaluated for its elastase inhibition activity; the authors proved that this extract have a strong inhibitory potential with a value of 89.64 ± 0.9% at (100 μg/mL). Moreover, the authors purified two active compounds, identified as quercetin-3-O-rhamnoside and myricetin-3-O-rhamnoside.

Many other investigations have described the enzyme inhibitory potential of *Pistacia* species [55]. However, no data have been found on the collagenase inhibitory potential of *Pistacia lentiscus*. Additionally, no studies have reported the anti-collagenase and anti-elastase activities of the genera *Halimium* and *Adenocarpus*. Only two articles have reported the collagenase and elastase activities of two plants belonging to the fabaceae family. In the first report, Chattuwatthana and Okello [56] mentioned the anti-collagenase and anti-elastase activities of *Pueraria candollei*; according to these authors, this plant would have a weak anti-collagenase activity and an anti-elastase activity with an IC50 of 143.0 μg/mL. The second study carried out on Cassia sp. described a weak anti-collagenase potential and an anti-elastase activity with an IC50 of 582.30 μg/mL [55].

Long exposure to harmful stimuli, such as ultraviolet (UV) radiation, activates the overexpression of collagenase and triggers a chain reaction that leads to the degradation of the skin extracellular matrix, causing skin aging [57]. Polyphenols play many roles at intra- and extracellular levels. They protect cells from damage caused by UV radiation and inhibit the degenerative activity of collagenase and elastase. It can be assumed that the enzyme inhibitory activity of the methanolic extracts of *Pistacia lentiscus* and *Halimium antiatlanticum* may be due to the abundance of polyphenols and flavonoids on these extracts. On the other hand, the elastase inhibitory activity may be due to alkaloids or their synergetic activity with other phytoconstituents. As a matter of fact, many bibliographic resources reported the elastase inhibitory activity of polyphenols as well as of alkaloids [54,58,59].

Currently, the cosmetic industry is interested in natural products with elastase, collagenase and melanogenesis inhibitory activities. Such natural compounds are mainly used as active agents in anti-wrinkle, anti-aging, skin lightening and sunlight protection products [60]. An increased demand for natural substances with these properties gave rise to several pharmacological investigations, including the current one.

## 4. Conclusions

Our studies showed that *Pistacia lentiscus* and *Halimium antiatlanticum* had the highest value of extraction yield, total phenolic and flavonoid contents and strong antioxidant and collagenase inhibitory activities, whereas *Adenocarpus artemisiifolius* had more alkaloids and a stronger elastase inhibition potential.

These results suggests that *Pistacia lentiscus*, *Halimium antiatlanticum* and *Adenocarpus artemisiifolius* have huge cosmetic potential due to their interesting phenolic and alkaloidic compositions, in addition to their antioxidant and skin-aging-related enzyme inhibitory potential.

This study is a part of the overall vision to fully study the phytochemistry and bioactivity of Moroccan endemic plants that have not been studied before. This paper is the first study to report *Halimium antiatlanticum* and *Adenocarpus artemisiifolius* in terms of the total phenolic and flavonoid contents and their antioxidant and dermocosmetic activities. In conclusion, our results provide evidence that Moroccan *Halimium antiatlanticum* and *Adenocarpus artemisiifolius* possess active substances that may be responsible for biological activities (tannins, flavonoids, terpenoids, carotenoids and saponins). All the studied species contained a considerable amount of polyphenols; we also found a correlation between phenolic content and the antioxidant capacity. In the present study, the analysis of free radical scavenging activities showed that *Halimium antiatlanticum* and *Adenocarpus artemisiifolius can* be a potent source of natural antioxidants. Moreover, *Halimium antiatlanticum* extract is a good inhibitor of collagenase, whereas *Adenocarpus artemisiifolius*

extract inhibited the elastase enzyme activity. These plants could be used as a natural antioxidant and preservative in food, non-food systems and active agents in cosmeceuticals. Further phytochemical analyses are required to isolate the elements of the plant responsible of these pharmacological activities.

**Author Contributions:** Conceptualization, N.E.A.; methodology, H.M.; validation, N.E.A., M.E.Y. and A.E.H.; formal analysis, J.M.R.; investigation, N.E.A., H.M. and S.H.; resources, N.E.A.; data curation, J.M.R.; writing—original draft preparation, H.M.; writing—review and editing, N.E.A., J.R.G.d.S.A. and J.M.R.; supervision, N.E.A. and F.M.; project administration, N.E.A.; funding acquisition, N.E.A. and J.M.R. All authors have read and agreed to the published version of the manuscript.

**Funding:** This research was partially funded by EXANDAS-H2020-MSCA-RISE-2015—"Exploitation of aromatic plants' by-products for the development of novel cosmeceuticals and food Supplements" (Grant Agreement No 691247).

**Institutional Review Board Statement:** Not applicable.

**Informed Consent Statement:** Not applicable.

**Data Availability Statement:** Data are contained within the article.

**Acknowledgments:** This work is also based upon the work from COST Action 18101 SOURDOMICS— *Sourdough biotechnology network towards novel, healthier and sustainable food and bioprocesses* (https://sourdomics.com/; https://www.cost.eu/actions/CA18101/ accessed on 14 June 2022), where the author J.M.R. is the Chair and Grant Holder Scientific Representative, and the author N.E.A. is member, and is supported by COST (European Cooperation in Science and Technology) (https://www.cost.eu/ accessed on 14 June 2022). COST is a funding agency for research and innovation networks. Regarding the author J.M.R., he was also financially supported by LA/P/0045/2020 (ALiCE) and UIDB/00511/2020-UIDP/00511/2020 (LEPABE) funded by National funds through FCT/MCTES (PIDDAC).

**Conflicts of Interest:** The authors declare no conflict of interest.

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
