# Peer review of "Phytochemical Screening, and In Vitro Evaluation of the Antioxidant and Dermocosmetic Activities of Four Moroccan Plants: Halimium antiatlanticum, Adenocarpus artemisiifolius, Pistacia lentiscus and Leonotis nepetifolia"

_cosmetics, doi:10.3390/cosmetics9050094_

Round 1

Reviewer 1 Report

The study reports the evaluation of some plants for their phytoconstituents and biological activities. The phytochemical screening as well as the total phenolic and flavonoid content were determined on the methanolic extract of these plants and subsequently the antioxidant and dermocometic activities were also evaluated. The results reported by the authors highlight the presence both in the leaves and in the aerial parts of tannins, polyphenols, flavones, coumarins, carotenoids, terpenoides and saponins. The polyphenol content is higher in Pistacia lentiscus and Halimium antiatlanticum and therefore also the antioxidant activity. The work on the whole is a good one, even with results of some importance and with developments of some interest for the cosmetic industry. In my opinion, due to its content and the clarity the presentation, the manuscript can be taken into consideration for publication on Cosmetics.

Author Response

we improve the english as required

Reviewer 2 Report

Manuscript number cosmetics-1888311

entitled: Phytochemical screening, and in-vitro evaluation of the antioxidant and dermocosmetic activities of four Moroccan plants: Halimium antiatlanticum, Adenocarpus artemisiifolius, Pistacia lentiscus and Leonotis nepetifolia

 This is a valuable and well-conducted scientific study, done thoroughly and expressed concisely. Therefore, the manuscript is suitable for Cosmetics after considering the below comments:

 1.      Abstract. Typos “ soxhlet®, change into “soxhlet apparatus.

2.      Page 3, line 117, please add space “at 40space°C,” „at 4space°C”, and passim.

3.      Typos. Page 5, line 208 „Potassium ferricyanide” change into “potassium ferricyanide”, ), change to ).

4.      Authors often use a capital letter unnecessarily, e.g., on page 10, line 427, “The Ethanolic”, page 12, line 510, “Methanolic.”

5.      Please use the MDPI style, e.g., journal abbreviations, not full journal names, e.g., Ref. 1 “European journal of medicinal chemistry.”

Author Response

  •  Soxhlet® replaced by Soxhlet apparatus : Line 23
  • Addition of space between number and unit : lines 108, 226, 273, 300, 306, 307
  • Potassium ferricyanide replaced by potassium ferricyanide  : line 204
  • Ethanolic replaced by ethanolic : line 428
  • Methanolic replaced by methanolic : line 512
  • Name of journals in references replaced by their initials : lines 594 to 731
  • DOI of articles had been added: lines 594 to 731

Round 2

Reviewer 2 Report

The authors conducted important data. This is an interesting paper. The present version is much better therefore, the manuscript is suitable for publication in its present form.